# The Effect of Acute Erythromycin Exposure on the Swimming Ability of Zebrafish (*Danio rerio*) and Medaka (*Oryzias latipes*)

**DOI:** 10.3390/ijerph17103389

**Published:** 2020-05-13

**Authors:** Yanyi Li, Jiabo Zhang

**Affiliations:** College of Fisheries, Huazhong Agricultural University, Wuhan 430000, China; lyy773981695@163.com

**Keywords:** erythromycin, swimming ability, mRNA, *Oryzias latipes*, *Danio rerio*

## Abstract

Erythromycin is a widely used antibiotic, and erythromycin contamination may pose a threat to aquatic organisms. However, little is known about the adverse effects of erythromycin on swimming ability. To quantify erythromycin-induced damage to fish swimming ability, *Oryzias latipes* and *Danio rerio* were acutely exposed to erythromycin. The swimming ability of the experimental fish was measured after exposure to varying doses of erythromycin (2 µg/L, 20 µg/L, 200 µg/L, and 2 mg/L) for 96 h. Burst speed (U_burst_) and critical swimming speed (U_crit_) of experimental fish significantly decreased. In addition, gene expression analysis of *O. latipes* and *D. rerio* under erythromycin treatment (2 mg/L) showed that the expression of genes related to energy metabolism in the muscle was significantly reduced in both species of fish. However, the gene expression pattern in the head of the two species was differentially impacted; *D. rerio* showed endocrine disruption, while phototransduction was impacted in *O. latipes*. The results of our study may be used as a reference to control erythromycin pollution in natural rivers.

## 1. Introduction

Antibiotics are frequently used worldwide for sterilization and to maintain health [1,2]. It has been reported that 80% of antibiotics enter the aquatic environment in their original form [3,4]. Moreover, studies have shown that antibiotics are widely detected in aquatic systems [5]. Antibiotics are considered to be a pollutant with a sustained adverse effect on the ecological environment [6]. Adverse effects on fish have been widely reported; for example, antibiotics have been shown to delay the hatching of fish eggs [7], damage gills and liver [8], and destroy the antioxidant defenses in muscle, which affects fish metabolism [1,9] and injures neurons [1,9]. Antibiotics are usually detected in rivers and wastewater at the level of ug/L [3]. However, previous studies have found concentrations of antibiotics at mg/L levels in swine wastewater [10,11,12], and erythromycin is commonly used for livestock farming [13].

Erythromycin, a semisynthetic antibiotic bacteriostatic, has been a widely used antibiotic since the 1950s [14]. Sewage treatment systems do not efficiently dispose of organic pollutants; thus, erythromycin is ubiquitous in the aquatic environment [15]. It takes a long time for erythromycin to degrade naturally [16]; in addition, previous studies have demonstrated that erythromycin has adverse effects on fish [16,17,18]. However, these studies focused on the physiological impacts on fish, rather than the negative effects on behavior [19].

The swimming behavior of fish is closely related to all the life activities of fish. Burst speed (U_burst_) and critical swimming speed (U_crit_) are important aspects of swimming ability because they play significant roles in the life activities of fish. U_burst_ is vitally important for activities such as eating, avoiding predators, and competitive interaction [20,21], while U_crit_ may be critical for seasonal behaviors associated with migration and reproduction. In addition, U_crit_ and U_burst_ represent the aerobic and anaerobic capacity of fish, respectively. Red muscle contains hemoglobin, myoglobin, and mitochondria, and is connected to the vascular system. This muscle is thought to have a metabolic function associated with aerobic exercise. White muscle provides a strong but limited burst of movement [22]. The movement of fish depends directly on the energy expenditure of muscles. In addition, the nervous system in the brain is stimulated by environmental pollutants, which can also lead to abnormal behavioral patterns and swimming activity [23]. Swimming patterns may be affected by gene expression, which is obviously regulated by the stress response of fish to erythromycin.

In this experiment, *Oryzias latipes* and *Danio rerio* were exposed to different concentrations of erythromycin (2 µg/L, 20 µg/L, 200 µg/L, and 2 mg/L) for 96 h, and the U_crit_ and U_burst_ of each treatment group were compared. Based on the swimming ability results, the gene expressions of *O. latipes* and *D. rerio* treated with erythromycin (2 mg/L) were analyzed. According to the results of gene expression, the expression of selected genes in each treatment group was verified by qRT-PCR. This study provides a factual basis for studying the effects of erythromycin on fish gene expression and swimming ability.

## 2. Materials and Methods

### 2.1. Experimental Fish

*D. rerio* and *O. latipes* are classic model species. Four-month-old *D. rerio* and 6-month-old Singaporean *O. latipes* were obtained from Shanghai Feixi Biotechnology Co., Ltd. (Shanghai, China). The average weight (± SD) and average length (± SD) of the *D. rerio* used in this study were 0.56 ± 0.08 g and 2.83 ± 0.10 cm, respectively. *O. latipes* had an average weight of 0.28 ± 0.03 g and an average fork length of 3.23 ± 0.05 cm. After the experimental fish were transported to the laboratory, they were put into a plexiglass tank (Figure 1) (25 cm long, 25 cm wide, 35 cm deep) filled with tap water aerated in advance. In addition, these fish were temporarily kept in the tank to eliminate the stress of the transport process. During this period, the fish were kept under a 14 L/10 D photoperiod, and they were hand-fed a commercial diet (Chengdu, China) containing > 40% protein and > 7% lipids. Moreover, the water temperature, dissolved oxygen and pH were maintained at 26.5 ± 2℃, 99.2 ± 0.3% and 7.7 ± 0.2, respectively. Approximately one-third of the water in the tank was replaced twice per day with pre-aerated tap water.

### 2.2. Experimental Design

*D. rerio* and *O. latipes* were exposed to different concentrations of erythromycin (Sigma Aldrich, CAS: 114-07-8) at 0 µg/L, 2 µg/L, 20 µg/L, 200 µg/L, or 2 mg/L. The concentration gradient was chosen based on the following two considerations. The minimum concentration was based on the concentration at which antibiotics are usually detected. The highest concentration was the concentration that may cause obvious changes in swimming ability based on the results of preliminary experiments. All fish were fed twice per day in line with the adaptation period. The erythromycin solution in the water tank was replaced once per day to prevent changes in concentration due to photolysis and evaporation. After 96 h of exposure, experimental fish from each treatment group were randomly selected to measure U_crit_ or U_burst_. Moreover, the mRNA of muscle and head was extracted from the experimental fish (control group and 2 mg/L treatment group). Each test concentration and the control were performed in triplicate.

### 2.3. Measurement of U_crit_ and U_burst_

The equipment used to measure fish swimming ability was a medium-sized swimming tank (SW10150) produced by Loligo Systems (Denmark). The volume of the sealing part of the sink was 30 L. The test area specifications were 55 cm x 14 cm x 14 cm, and the flow rate of the test area ranged from 5 to 175 cm/s. The flow rate in the sealed sink was generated by the rotation of a motor, which could be changed by adjusting the drive, while the regulator was connected by the inverter and the cellular stabilizer on the left side of the test area to produce a uniform and constant flow field. In addition, a YSI Ecosense D0200A dissolved oxygen meter, digital flow speedometer (AC10000) and 30 mm vane wheel flow probe (AC10002) were used.

Before the start of the test, the dissolved oxygen level and temperature conditions in the device were checked, and the motor was gradually turned to remove bubbles from the test device. Then, a *D. rerio* was placed in the test area of the swimming device, and the test device was sealed. The flow rate was adjusted to 10 cm/s and then increased by 25 cm/s at 20 min intervals. For *O. latipes*, the flow rate was increased to 15 cm/s over 20 min and then increased by 10 cm/s every 20 min. Finally, an experimental fish swimming fatigue net was used to end the test. The following formula was used to calculate U_crit_:(1)$$U_{crit}=v_{2}+\frac{t_{2}}{\Delta t_{2}}\Delta v_{2}$$
where

$\Delta t_{2}$(min) is the duration of each flow rate, $t_{2}$(min) is the time the test fish remained at this flow rate, $\Delta v_{2}$(cm/s) is the velocity increment, and $v_{2}$(cm/s) is the maximum swimming speed of the test fish that was reached during
$\Delta v_{2}$.

At the end of the experiment, the flow rate was decreased, the fatigued test fish were removed, and their body length, weight and conventional morphological parameters were measured. During the test, if the dissolved oxygen concentration in the sink was less than 7 mg/L, the water in the sealed sink was exchanged with a water pump.

U_burst_ was also calculated according to the “incremental flow rate method”. A fish was placed in the test segment before the test and was adapted to a low flow rate (10 cm/s) for 20 min to eliminate the stress of the transfer process on the fish. After the test started, the flow rate in the test segment was gradually increased by 1 cm/s. When the test fish were fatigued and could not continue to swim, the test was stopped. The critical swimming ability and burst swimming ability tests were repeated three times.

### 2.4. RNA Sequencing

Experimental fish from the control group and the 2 mg/L treatment group were randomly selected and placed in MS222 at a concentration of 0.2 mg/L until anesthetized. The tail muscle tissue and head tissue were collected, processed and immediately placed in liquid nitrogen. The processed samples were then sent to Nanjing Personal Gene Technology Company for RNA determination and sequencing.

Total RNA was isolated with a RNeasy Mini Kit (Qiagen, Germantown, MD, USA). Then, additional DNase I (Qiagen) was added to digest contaminating genomic DNA. One microgram of integrated RNA per sample was prepared and sequenced using an RNA-Seq library. The mRNA library was built by the TruSeq RNA Sample Preparation Kit (Illumina, San Diego, CA, USA) with reference to the manufacturer’s instructions. The prepared mRNA samples were then clustered on an Illumina HiSeq 2500 for sequencing. The sequences from each treatment group were evaluated after 100 cycles. The RNA-Seq reads were assessed for quality control with FastQC (version 0.10.1; Babraham Bioinformatics, Cambridge, UK). All reads were evaluated by Bridger (r2014-12-01) (-pair_gap_length 50-min_kmer_coverage 4-min_ratio_non_error 0.15). The transcripts scored per million fragments per thousand bases of external subfragments mapped (RSM) were calculated according to Trinity script variance expression (false discovery rate (FDR) ≤ 0.05) using the blind dispersion method and Cuffdiff analysis, which yielded lists of upregulated and downregulated genes. Fisher’s exact test with FDR correction (FDR ≤ 0.05) was used to analyze gene functions and pathways by Gene Ontology (GO) functions and Kyoto Encyclopedia of Genes and Genomes (KEGG) pathways, respectively. The ratio of differentially expressed genes (DEGs) to the total number of genes in the associated pathways was considered an enrichment factor.

### 2.5. Quantitative Real-Time PCR (qRT-PCR) Validation

qRT-PCR was used to verify the results of RNA-Seq. Based on their position, function and expression level in the genome, eight differentially expressed transcripts were selected (6 upregulated and 2 downregulated). mRNA transcripts were aliquoted using a RealPlex4S qRT-PCR system (Eppendorf, Germany). RNA samples were extracted using an miRNeasy Mini Kit (50) (Qiagen). The final volume of the RT-PCR reaction was 25 µL. The thermocycler settings were as follows: 95 °C, 2 min; and 40 cycles of 95 °C for 10 s, 68 °C for 30 s, and 68 °C for 5 min. The relative expression level was calculated using the 2^−ΔΔCT^ method with β-actin as a reference gene. Three independent samples were analyzed in triplicate.

### 2.6. Statistical Analyses

The U_crit_ and U_burst_ of experimental fish were analyzed with Origin 8.0 software (Origin Lab Corporation, Northampton, MA, USA) and SPSS Statistics 20 (SPSS Inc., Chicago, IL, USA). Significant differences between the treatment groups were determined with one-way analysis of variance (ANOVA), and the significance level was set at *p* < 0.05. Transcriptomic data were analyzed and visualized by R software (Core Team, 2014).

### 2.7. Ethics Statement

The animal study proposal was approved by the Ethics Committee for Animal Experiments of Sichuan University (ethics code is 2019062101). All experimental procedures were performed in accordance with the Regulations for the Administration of Affairs Concerning Experimental Animals approved by the State Council of the People’s Republic of China.

## 3. Results

### 3.1. Swimming Performance and Muscle Fibers

The U_crit_ of *D. rerio* and *O. latipes* after 96 h of exposure to varying doses of erythromycin (2 µg/L, 20 µg/L, 200 µg/L, and 2 mg/L) is illustrated in Figure 2a,c. The U_crit_ of *D. rerio* was decreased by a small margin in the low-concentration treatment groups (2 µg/L, 20 µg/L and 200 µg/L). In addition, the U_crit_ of *D. rerio* decreased markedly to 14.17 BL/s in the high-concentration treatment group (2 mg/L) compared with the control group (26.32 BL/s) (U_crit_: df = 3, F = 5.32, *p* = 0.03); the lowest dose of erythromycin (2 µg/L) slightly decreased the speed of *O. latipes* to 15.25 BL/s, and the highest dose of erythromycin (2 mg/L) clearly reduced the U_crit_ value of *O. latipes* to 10.32 BL/s, while the U_crit_ of the control fish was 15.46 BL/s.

The U_burst_ of *D. rerio* and *O. latipes* is shown in Figure 2b,d. Although the U_burst_ of *D. rerio* improved slightly under treatment with low doses of erythromycin (2 µg/L and 20 µg/L), there was a striking overall downward trend with increased doses of erythromycin. The U_burst_ of *D. rerio* (20.12 BL/s and 16.57 BL/s) was dramatically lower in the 200 µg/L and 2 mg/L group than in the control group (27.12 BL/s) (U_burst_: df = 3, F = 2.45, *p* = 0.03) (U_burst_: df = 3, F = 4.28, *p* = 0.04); The U_burst_ of *O. latipes* fluctuated with increasing doses of erythromycin from 2 µg/L to 200 µg/L, but the U_burst_ value (14.23 BL/s) was notably lower in the 2 mg/L group than in the nonexposed group (19.84 BL/s) (U_burst_: df = 4, F = 3.65, *p* = 0.02).

The appearance of the exposed fish and the appearance of the control group were not observed to be different. the size of the muscle fibers changed significantly, as shown in Figure 3. The muscle fibers of *D. rerio* and *O. latipes* became thinner and weaker. Additionally, the pores of the muscle fiber bundle also became larger following erythromycin-induced stress.

### 3.2. mRNA Expression Levels of Genes

Deep sequencing data were analyzed for each obtained sample from the treatment groups. We considered read quality scores above Q30 (correct base recognition rate greater than 99.9%) to indicate clean reads, and the muscle gene expression results of the experimental fish showed that more than 94.2% of the *D. rerio* reads were clean, while more than 93.4% of the *O. latipes* reads were clean. The proportion of mapped reads in *D. rerio* sequences was higher than 97.86%, and was higher than 94.09% in *O. latipes.* With erythromycin treatment, there were 503 upregulated genes and 541 downregulated genes in *O. latipes* muscle, while 463 upregulated and 319 downregulated genes were found in the muscle of *D. rerio*, as shown in Figure 3.

To gain a better understanding of the effects of erythromycin on genes in *D. rerio* muscles, we further analyzed the KEGG pathways. Based on the KEGG enrichment analysis of the expressed genes, the top 20 pathways with the smallest *p*-values, indicating the most significant enrichment, were selected for presentation, as shown in Figure 4. The most significant KEGG pathway in *D. rerio* and *O. latipes* was oxidative phosphorylation. Specifically, *D. rerio* had 47 downregulated genes (cox6a2, ndufs6, cox5b2 etc.), and *O. latipes* had 53 downregulated genes (ndufs4, ndufa12, ndufb3 etc.).

The gene expression profiles of the heads of the experimental fish showed that *D. rerio* had more than 93.13% clean reads and *O. latipes* had more than 92.7%. The proportion of mapped reads in the *D. rerio* sequences was higher than 95.1%, and the proportion of mapped reads in *O. latipes* was more than 94.09%. Genes in the fish head were impacted by erythromycin treatment, and there were 342 upregulated genes and 1106 downregulated genes in the head of *O. latipes*. For *D. rerio*, 461 genes were upregulated, and 551 genes were downregulated, as shown in Figure 5.

To obtain insight into the alteration in gene expression in the head of *D. rerio* induced by erythromycin, KEGG pathways were analyzed. We selected the top 20 pathways with the smallest *p*-values and the most significant enrichment for presentation based on KEGG pathway enrichment analysis, as presented in Figure 6. For *O. latipes*, the KEGG analysis results showed that the two most significant pathways in organismal systems were the adipocytokine signaling pathway (npy and ppara) and the PPAR signaling pathway (lpl). In addition, for *D. rerio*, the most significant pathway in the organismal systems category was phototransduction (guca1a, grk7b and grk1a).

### 3.3. Validation of RNA-Seq DEG Expression Profiles in Danio rerio and Oryzias latipes by qRT-PCR

Fifteen DEGs in the RNA-Seq results were selected for expression pattern verification by qRT-PCR using cDNA from the remaining RNA samples from the different erythromycin treatment groups. These 15 genes were identified in the *D. rerio* head (4 DEGs) and muscle (4 DEGs) or in the *O. latipes* head (3 DEGs) and muscle (4 DEGs). The 15 genes reported in the RNA-Seq data (2 upregulated and 13 downregulated) are shown in Figure 7. All of these genes were significantly changed in the high-concentration exposure group compared with the control group because the adverse effects were amplified. However, at low concentrations, these genes had unstable expression patterns. Nevertheless, all of these genes were significantly changed in the same direction as in the other groups, which verified the results of RNA-Seq (Figure 8).

## 4. Discussion

### 4.1. Swimming Performance

Antibiotics had a greater negative effect on the U_crit_ of *D. rerio* than on that of *O. latipes*. The U_crit_ of both species was significantly impaired at high exposure concentrations. Other environmental pollutants have been shown to reduce the swimming ability of other species of fish. For example, the U_crit_ of juvenile Florida pompano was significantly reduced from 90.10 ± 1.35 cm/s to 84.20 ± 1.36 cm/s under the toxic influence of methanol [24]. Another study found that U_crit_ of *Erimyzon sucetta* was reduced by approximately 50% when they were exposed to ash [25]. Prolonged swimming activities (such as U_crit_) may be sensitive to changes in maximum aerobic capacity, cardiac output, muscle fiber function, and anaerobic metabolism. The results showed that the aerobic capacity of the fish was impaired. There are several possible reasons for this impairment. Fish are poisoned by environmental pollutants and are forced to detoxify a large amount of oxygen [26,27], which reduces the oxygen supplied for exercise [27]. In addition, water pollution can change the shape of gill tissue, resulting in an impaired tissue oxygen supply during exercise [28,29]. Moreover, sublethal exposure to contaminating elements can also lead to increased hemoglobin and plasma protein concentrations, resulting in increased blood concentrations and local tissue hypoxia [30].

The U_burst_ of *D. rerio* and *O. latipes* showed decreases of 38% and 31%, respectively, under treatment with 2 mg/L erythromycin. Explosive activity occurs rapidly in a short period of time, and burst swimming performance is mainly affected by anaerobic metabolism; thus, some functions related to anaerobic metabolism in both species may be hindered. U_burst_ reflects the ability of a fish to perform short-term anaerobic movements while foraging or avoiding danger [31]. This result reveals that the ability of experimental fish to hunt and evade predators was weakened, which may reduce the survival rate of experimental fish in the long run. Previous studies have also found that toxic contamination can adversely affect the U_burst_ of fish. For instance, the U_burst_ of ash-exposed *E. sucetta* was decreased by 30%–104% compared to that of control fish [25,26,27].

The swimming performances of *D. rerio* and *O. latipes* were weakened by 2 mg/L erythromycin. Our results indicate that the weakening of the muscle fibers of the experimental fish is a possible reason for the decline in swimming ability. Previous studies on toxin exposure can lead to body dysfunction and muscle structure [32]. Under the treatment of contaminants, zebrafish somatic muscle fibers affect the arrangement or integrity of muscles, and the abnormality of muscles directly affects exercise ability. Specifically, the impairment of myogenesis and the destruction of myofilament tissue caused by contaminants impedes the ability of muscles to contract and thus significantly reduces swimming performance of zebrafish [33]. Another possible reason is that aerobic and anaerobic metabolism is regulated by different physiological mechanisms, and the results of the swimming ability assay showed that erythromycin had adverse effects on these two physiological mechanisms. The effect of other antibiotics on fish mobility also supports these results. Studies showed that high concentrations of antibiotics, even for short periods of time, can induce behavioral disturbances in fish [34]. Both aerobic and anaerobic metabolic capacities were inhibited by erythromycin exposure. This suggests that erythromycin affects the energy metabolism of fish.

### 4.2. Analysis of mRNA in Muscle

Swimming behavior is closely related to muscle contraction. A decline in swimming ability reflects a decline in muscle function. Moreover, aerobic metabolism involves the transport of oxygen and carbohydrates through respiration and circulation. It reflects the metabolic processes throughout the organism, from skin to muscle tissue, which may affect the absorption and transport of oxygen. Additionally, aerobic activity requires a continuous supply of adenosine triphosphate (ATP) from fish in different organs and muscles [34,35]. Anaerobic movement is a temporary explosion with a limited range of rapid metabolism [36]. It allows carbohydrates and oxygen to enter the muscle, which consumes glycogen and phosphocreatine during explosive movement. The power of anaerobic activity depends on ATP in the absence of oxygen [37,38]. Both types of exercise depend on the ability of muscles to produce and release ATP. The mRNA expression results of the muscles from both species confirm their behavior and further explain the decline in swimming ability.

The markedly enriched KEGG pathway shown in Figure 4 was oxidative phosphorylation. Additionally, both species had downregulated genes in this pathway, which suggests that the energy metabolism of *D. rerio* and *O. latipe**s* was inhibited by erythromycin. Oxidative phosphorylation is a metabolic pathway in cells. This process occurs in the mitochondrial inner membrane of eukaryotes or in the cell membrane of prokaryotes and uses the enzymes and the energy released by the oxidation of various nutrients to synthesize ATP. ATP is the primary molecule that directly provides energy to anabolic organisms. For most aerobic organisms, the tricarboxylic acid cycle-oxidative phosphorylation is the main process that produces ATP. The first four significantly altered genes in this pathway in *D. rerio* and *O. latipes* were validated. In addition, these genes encode NADH dehydrogenase, cytochrome c oxidases, and F-type ATPase (eukaryotes). NADH dehydrogenase is used in the electron transport chain to generate ATP. It is a receptor oxidoreductase that catalyzes the following chemical reactions: NADH + H+ + acceptor ⇌ NAD+ + reduced acceptor [39]. Cytochrome c oxidases have beneficial effects on exercise, such as increasing oxygen levels in vascular tissues. Enzymes cannot reduce oxygen, resulting in oxygen accumulation and the diffusion of oxygen into surrounding tissues [40]. The downregulation of this gene means the loss of these beneficial effects. Another study suggested that the suppression of cytochrome c oxidases decreases the rate of cellular respiration [41]. F-ATPase, also known as F-Type ATPase, is involved in many basic cellular metabolic activities (such as acidosis, alkalosis and respiratory gas exchange). The gene expression results provide an explanation for the decrease in the swimming ability of *D. rerio* and *O. latipes* under erythromycin stress. Antibiotics adversely affected the energy metabolism pathways of these two fish. Specifically, ATP synthesis and ATP release were inhibited, which inevitably damaged their swimming ability.

### 4.3. Analysis of mRNA in Head

Muscle energy supply is directly related to swimming behavior, and the central nervous system can also indirectly negatively impact fish swimming ability. Abnormal functions related to the central nervous system may cause sensory organ dysfunction and movement disorders. It can also lead to hormone disorders that block energy metabolism. The sensitivity of fish to stress induces changes in behavior, and the fields of behavioral ecology and toxicology provide a clear explanation of this connection. Biochemical disorders, such as neurotransmitter and thyroid changes, affect the behavior of fish [42]. *D. rerio* phototransduction was subdued under the pressure of erythromycin. Phototransduction is the conversion of the distribution and wavelength of photons into neuron activity patterns, which then induce movement and endocrine responses. In *D. rerio* and in mice with mutations in a light-sensing gene, researchers found both motor and motor coordination mutations [43,44]. The optokinetic response (OKR) requires the retina as a photosensitive organ and employs motor thrust, as does swimming [43,44]. Mutant *D. rerio* have slow eye movement and impaired motor ability [45]. In addition to the extreme situation, a more common phototransduction function is to regulate circadian rhythm through visual pigments [46]. Impaired light transmission can disrupt the body clock, which reverses day and night and affects metabolism. The mRNA expression results from the head of *O. latipes* revealed neuroendocrine disruption under the pressure of erythromycin. Biological neuroendocrine systems are used to regulate food intake, metabolism, and energy distribution to ensure a steady supply of energy [46,47]. Metabolism and aerobic exercise are intertwined because of a common connection: they both depend on the intake of food, which is the source of chemical energy for these processes. In the short term, there is a match between energy intake and expenditure. In the long term, they are carefully balanced and regulated by several endocrine systems that work together to ensure energy homeostasis [48,49]. The hypothalamus is crucial for monitoring energy balance [50]. The hypothalamus regulates energy by signaling fat storage. Another theory is that the hypothalamus regulates energy balance by storing signals from carbohydrates [51].

The most affected biological system in *D. rerio* was phototransduction, a sensory transduction pathway in the visual system. Through this process, light is converted into electrical signals in the rods, cones, and photosensitive ganglion cells of the retina in the eye. In the dark, the main function of the downregulated genes is to control the transmission of Ca^2+^ ions, thereby inhibiting neurotransmitters. In light, the downregulated genes primarily affect the function of retinal porphyrin. The inactivation of rhodopsin may cause the loss of dark adaptation. These downregulated genes indicate that the vision of *D. rerio* is impaired in light and darkness, and that visual adaptability is damaged when light and dark alternate. The visual sensitivity of fish to light and dark requires much energy [52], but phototransduction is very important for fish to hunt and avoid danger. *D. rerio* had weakened visual ability under the pressure of erythromycin, which means that the chances of survival were reduced. In addition to erythromycin, other environmental pollutants may also cause the visual function of fish to be suppressed. For example, tributyltin has also been reported to block this pathway in fish [53].

For *O. latipes*, the upregulated genes related to a biological system were Neuropeptide Y (NPY) and PPAR-α. NPY is a 36 amino acid neuropeptide that participates in various physiological processes and in the homeostasis of the central nervous system and peripheral nervous system [54,55,56,57]. In the head, NPY is produced in different parts of the hypothalamus. In addition, NPY is thought to have multiple functions, including changing the storage of fat energy and reducing anxiety and stress [57,58]. NPY regulates the neuroendocrine release of various hypothalamic hormones, such as luteinizing hormone [59]. In particular, NPY is considered to be an endogenous anxiolytic peptide. The level of NPY can be regulated by stress, and it is considered necessary for stress regulation [60]. Higher levels of NPY may be self-regulating to alleviate fear responses [61]. The upregulation of NPY in *O. latipes* is a sign of fear and anxiety. In addition, there may be disorders in fat metabolism. PPAR is an important transcription factor. PPAR-α regulates many genes involved in various aspects of lipid metabolism. The upregulation of this gene may activate functions related to lipid metabolism. However, the downregulation of Lpl demonstrates that fatty acid transport and a component of lipid metabolism was inhibited [62]. That means erythromycin can cause many lipoprotein metabolism abnormalities.

## 5. Conclusions

The swimming ability of the *Oryzias latipes* and *Danio rerio* was measured after exposure to varying doses of erythromycin (2 µg/L, 20 µg/L, 200 µg/L, and 2 mg/L) for 96 h. U_burst_ and U_crit_ of the experimental fish did not change significantly at low concentrations(2 µg/L, 20 µg/L, 200 µg/L). The swimming ability of both *O. latipes* and *D. rerio* was reduced by exposure to high concentrations of erythromycin, and U_crit_ decreased to 53% and 71%, while U_burst_ decreased by 39% and 23%, respectively. This finding indicates that the aerobic capacity and anaerobic capacity of these fish were reduced. mRNA expression analysis of the muscle confirmed that ATP production- and ATP release-related functions were inhibited. Erythromycin has different effects on gene expression in the head of the two species, but the results from both species provide indirect evidence of behavioral changes. Phototransduction of *O. latipes* was inhibited, which may lead to abnormal behavior or body clock disorders. The hormone imbalance in *O. latipes* may lead to energy metabolism disorders, and its fat metabolism abnormalities.

## Figures and Tables

**Figure 1 ijerph-17-03389-f001:**
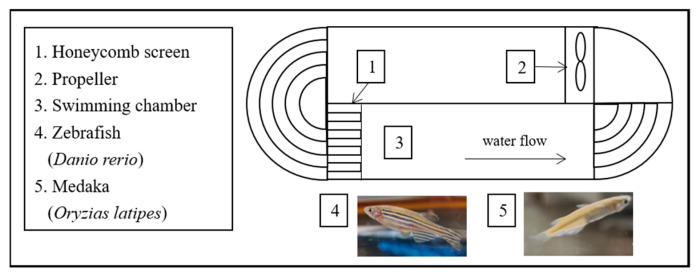
Sketch of the experimental setup in the laboratory.

**Figure 2 ijerph-17-03389-f002:**
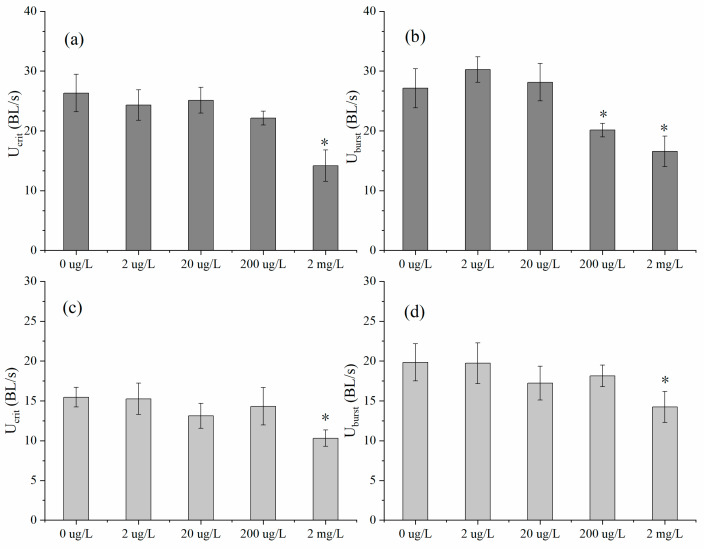
(**a**) Critical swimming speed (U_crit_) and (**b**) burst speed (U_burst_) of *Danio rerio*; (**c**) U_crit_ and (**d**) U_burst_ of *Oryzias latipes* exposed to varying concentrations (0 µg/L, 2 µg/L, 20 µg/L, 200 µg/L, and 2 mg/L). Significant differences between the control groups and erythromycin exposed groups, * repersents *p* < 0.05.

**Figure 3 ijerph-17-03389-f003:**
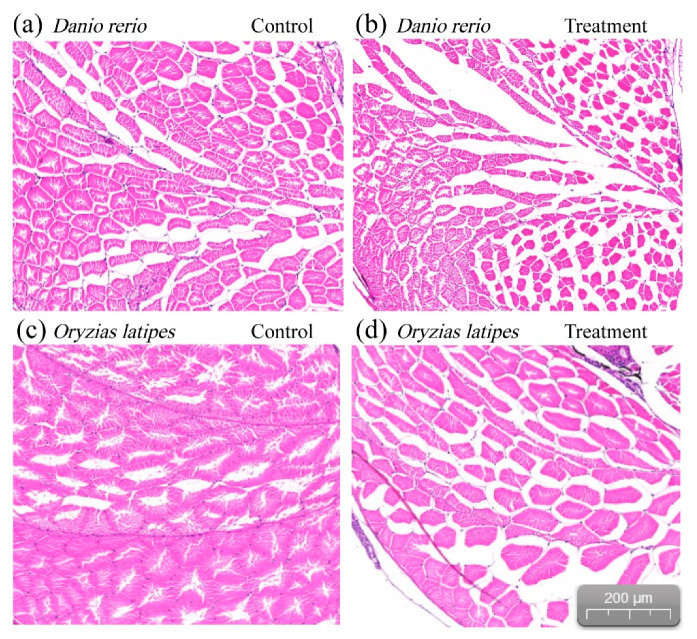
Thinner and weaker muscle fibers bundles exposed to 2 mg/L. (**a**) and (**b**) of *D. rerio*; (**c**) and (**d**) of *O. latipes*.

**Figure 4 ijerph-17-03389-f004:**
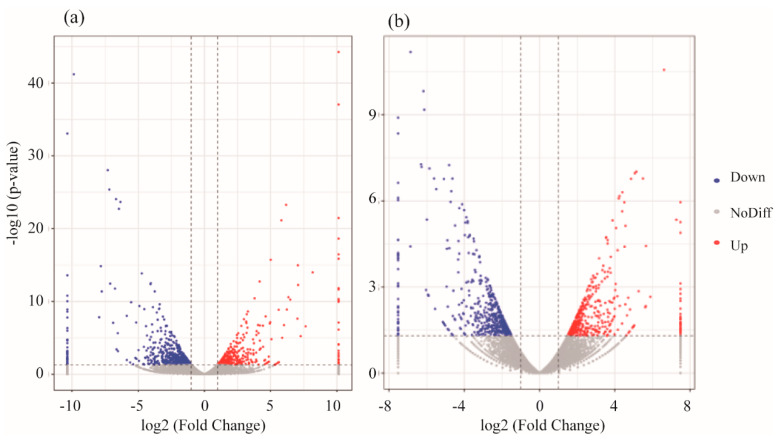
Volcano plots of the gene libraries of muscle from fish in the antibiotic-treated groups and control group showing the variance in gene expression with respect to FC and FDR. Each dot represents an individual gene: the black dots on the left represent the downregulated genes, and the red dots on the right represent the upregulated genes (**a**) *D. rerio*, (**b**) *O. latipes*.

**Figure 5 ijerph-17-03389-f005:**
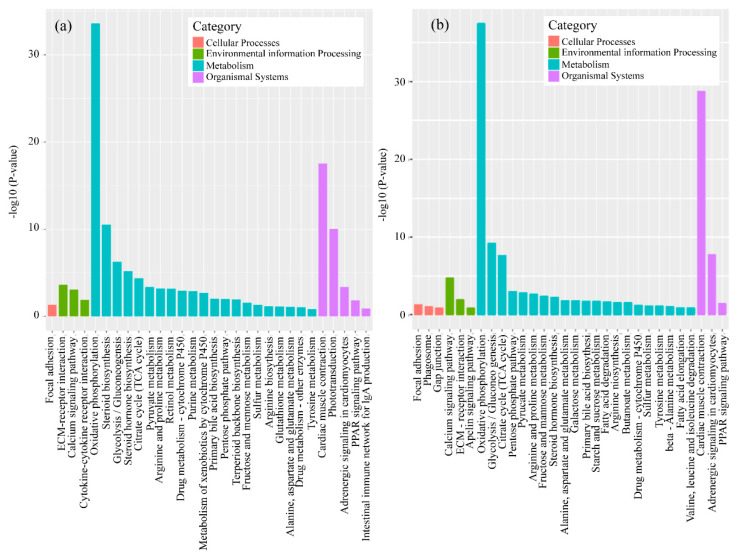
The top 20 enriched Kyoto Encyclopedia of Genes and Genomes (KEGG) pathways in the erythromycin-treated groups and the control group (**a**) *D. rerio*, (**b**) *O. latipes*.

**Figure 6 ijerph-17-03389-f006:**
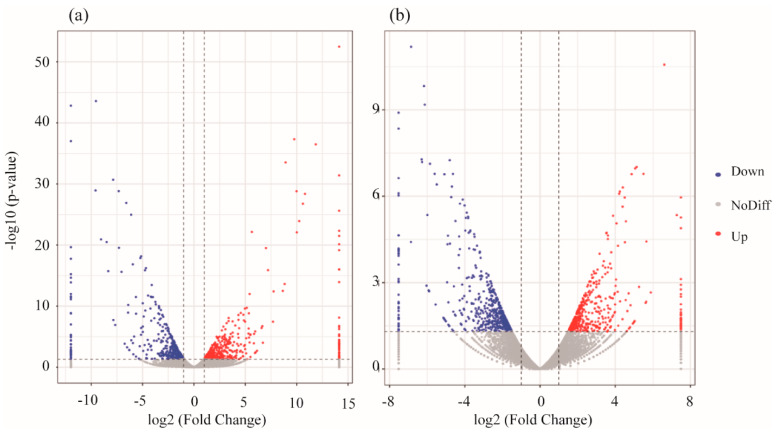
Volcano plots of the gene libraries of heads from experimental fish in the antibiotic-treated groups and control group showing the variance in gene expression with respect to FC and FDR. Each dot represents an individual gene: the black dots on the left represent the downregulated genes, and the red dots on the right represent the upregulated genes (**a**) *D. rerio*, (**b**) *O. latipes*.

**Figure 7 ijerph-17-03389-f007:**
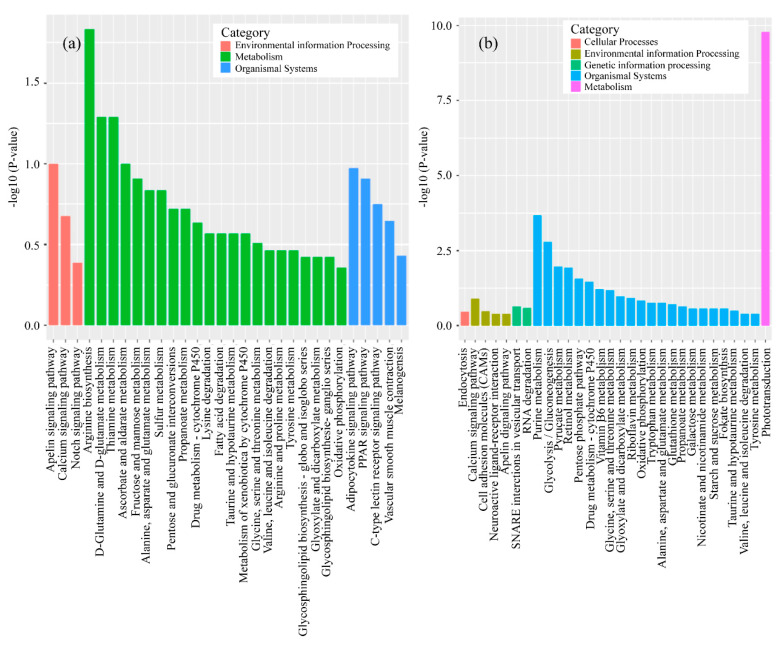
The top 20 enriched KEGG pathways in the erythromycin-treated groups and the control group (**a**) *D. rerio*, (**b**) *O. latipes*.

**Figure 8 ijerph-17-03389-f008:**
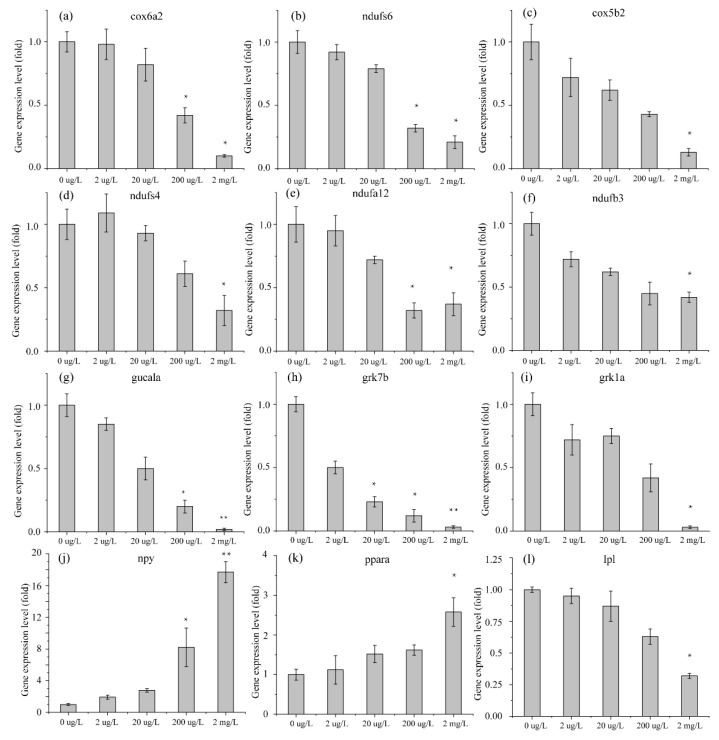
The effects of erythromycin on the mRNA expression of genes in *D. rerio* and *O. latipes*. ((**a**–**c**) gene *in D. rerio*’s muscle; (**d**–**f**) gene in *O. latipes*’s muscle; (**g**–**i**) gene in *D. rerio*’s head; (**j**–**l**) gene in *O. latipes*’s head) Values represent the mean ± SD (n = 3 replicates). * indicates *p* < 0.05 relative to the control.

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
