# Peer review of "The Effect of Acute Erythromycin Exposure on the Swimming Ability of Zebrafish (*Danio rerio*) and Medaka (*Oryzias latipes*)"

_ijerph, 2020, doi:10.3390/ijerph17103389_

Round 1
Reviewer 1 Report
The study investigates the effects of erythromycin exposure of zebrafish. Manuscript is well-written overall. I have the following suggestions:
- Authors should explain what is novel about their study. Line 37 seems to point this out. However, although they measure swimming speed, they connect this to expression of genes related to metabolism (not nervous system). Therefore, it seems like they are also measuring a physiological effect.
- Authors should comment on how realistic is to use a 30 L fish tank (size and other parameters) to model an ocean/sea/river fish environment.
- Figure 2 caption is not clear
- Figure 4 and 6 axes font is too small – cannot read them. When I enlarge the images, resolution is lost. They should either increase the resolution or make the fonts bigger.
- Looks like data shows no significant difference in fish swimming rated at relevant concentration. This should be added to the conclusions. It will be good if they can make suggestions for lowest allowable amount for this specific antibiotic.
- Line 386 has two extra words
Author Response
- Authors should explain what is novel about their study. Line 37 seems to point this out. However, although they measure swimming speed, they connect this to expression of genes related to metabolism (not nervous system). Therefore, it seems like they are also measuring a physiological effect.
Response 1: This article focuses on the adverse effects of antibiotics on swimming ability. Changes in mRNA expression were only concerned with functions related to swimming ability. The experiment was designed to explore possible reasons for the changes in swimming
- Authors should comment on how realistic is to use a 30 L fish tank (size and other parameters) to model an ocean/sea/river fish environment.
Response 2: To control the variables in a laboratory environment, adult fish were placed in a tank of still water during the experiment [1]. The dimensions of the tank are designed to take into account the dimensions of the experimental fish. Make sure the fish are free to swim and are not threatened by space constraints.
- Figure 2 caption is not clear
Response 3: Figure 2 caption has been modified on lines 175 ~ 177
- Figure 4 and 6 axes font is too small – cannot read them. When I enlarge the images, resolution is lost. They should either increase the resolution or make the fonts bigger.
Response 4: Thanks for your reminding that the illustration is not clear enough. The content of the picture is not clearly presented. We have revised the Figures to increase the readability.
- Looks like data shows no significant difference in fish swimming rated at relevant concentration. This should be added to the conclusions. It will be good if they can make suggestions for lowest allowable amount for this specific antibiotic.
Response 5: Thank you very much for the valuable comments of the reviewers. This part has been added to the conclusion in lines 370~373; The results of exposure of fish eggs to antibiotics indicate that the minimum concentrations that cause developmental, biochemical and behavioural disturbance of fish eggs are different [2]. Besides Studies on the stress of antibiotics in adult fish indicate differences in damage to different organs of adult fish [3]. Therefore, further research is needed to propose the lowest concentration that a given antibiotic environment can allow. Our research focuses only on the swimming ability of adult fish, and the reference significance of the minimum concentration proposed based on these results is limited.
- Line 386 has two extra words
Response 6: Fund has been updated
References:
[1] Zhao, Ye, et al. "The influence of three antibiotics on the growth, intestinal enzyme activities, and immune response of the juvenile sea cucumber Apostichopus japonicus selenka." Fish & shellfish immunology 84 (2019): 434-440.
[2] Andrade T S, Henriques J F, Almeida A R, et al. Carbendazim exposure induces developmental, biochemical and behavioural disturbance in zebrafish embryos[J]. Aquatic Toxicology, 2016, 170: 390-399.
[3] Chemello, Giulia, et al. "Safety assessment of antibiotic administration by magnetic nanoparticles in in vitro zebrafish liver and intestine cultures." Comparative Biochemistry and Physiology Part C: Toxicology & Pharmacology 224 (2019): 108559.
Reviewer 2 Report
The topic of the article is very interesting, however some adjustment to the experimental design and the conclusion extrapolated from the experiments have to be done.
in details:
In Figure 2 the swimming ability is significantly impaired only at the highest concentration used (2mg/L). that is one order of magnitude higher than the amount detected in the rivers (line 75-76-77 page2). Therefore, the authors should better explain the impact on river fishes of the erythromycin.
Why the authors choose adult fishes and not early stages of life when probably the swimming ability are more affected? Please explain.
In Fig.2 there are not small letters indicating the significant differences (line 175-176 page 5).
the experiments in Fig 3-4-5-6 are performed for the concentration of 2mg/L , is this concentration representative of the environmental reality? Also, for the gene expression in figure 7 most of the significant differences correspond to the highest treatment condition. Can the authors explain the results in the context of the pollution in natural rivers?
Moreover, regarding the effect on the lipid metabolism the authors should consider early stages of life. If the fishes are born and growth in erythromycin polluted water, they can or develop a sort of resistance or be lethally effect decrease the ability to reach the reproductive stage. Experiment in young fishes are highly recommended.
Author Response
- In Figure 2 the swimming ability is significantly impaired only at the highest concentration used (2mg/L). that is one order of magnitude higher than the amount detected in the rivers (line 75-76-77 page2). Therefore, the authors should better explain the impact on river fishes of the erythromycin.
Response 1: Although the antibiotic concentration is low in most water bodies, the antibiotic concentration is high in some natural water bodies due to the abuse of antibiotics. Tetracycline antibiotics are frequently found in wastewater of farms, and detected at mg/L concentrations [1,2]. However, some studies have investigated the occurrence of tetracycline antibiotics in swine waste-water and have observed high concentrations of chlortetracycline (3.99-107.14mg/L), oxytetracycline (0.67-5.65 mg/L) and tetracycline (0.13-1.78 mg/L) [3~5]. the lowest concentration of antibiotics based on those frequently detected in the river and the highest concentrations to potentially cause abnormal behavior to adult zebrafish based on our pre-liminary results.
- Why the authors choose adult fishes and not early stages of life when probably the swimming ability are more affected? Please explain.
Response 2: The behavioral changes of larvae subject to antibiotics are mainly concerned with the minimum distance, maximum distance and swimming time [6]. This article focuses on the burst speed (Uburst) and critical swimming speed (Ucrit) of fish. Uburst is vitally important for activities such as eating, avoiding predators, and competitive interaction, while Ucrit may be critical for seasonal behaviors associated with migration and reproduction. Therefore, these two swimming abilities are closely related to the life activities of adult fish while larvae is the absence of Ucrit and Uburst.
- In Fig.2 there are not small letters indicating the significant differences (line 175-176 page 5).
Response 3: Legend has been modified 175 ~177
- the experiments in Fig 3-4-5-6 are performed for the concentration of 2mg/L, is this concentration representative of the environmental reality? Also, for the gene expression in figure 7 most of the significant differences correspond to the highest treatment condition. Can the authors explain the results in the context of the pollution in natural rivers?
Response 4: The highest concentration (2 mg / L) used in this study was higher than the antibiotic concentration in most rivers. When investigating the principles of physiological and behavioral changes in the acute exposure of aquatic organisms to antibiotics, high concentrations of antibiotics are used to amplify the adverse effects of antibiotics as a common method of toxicology [7,8]. Long-time effects of erythromycin of natural river concentrations is our next experiment.
- Moreover, regarding the effect on the lipid metabolism the authors should consider early stages of life. If the fishes are born and growth in erythromycin polluted water, they can or develop a sort of resistance or be lethally effect decrease the ability to reach the reproductive stage. Experiment in young fishes are highly recommended.
Response 5: We mainly focus on changes in fish behavior. mRNA is mainly used to explain possible causes of behavioral changes. Results Analysis of lipid metabolism may be the cause of the weakening of swimming ability of fish. Because this study does not focus on lipid metabolism. Therefore, experiments were not designed from this perspective.
References:
[1] Ben, W., Qiang, Z., Adams, C., Zhang, H., Chen, L., 2008. Simultaneous determination of sulfonamides, tetracyclines and tiamulin in swine wastewater by solid-phase extraction and liquid chromatography-mass spectrometry. J. Chromatogr. A1202 (2), 173-180.
[2] Xu, Y.G., Yu, W.T., Ma, Q., Zhou, H., Jiang, C.M., 2015. The antibiotic in environment and its ecotoxicity: a review. Asian J. Ecotoxicol. 10 (3), 11-27.
[3] Lei, H., 2016. Occurrence, Fate and Environmental Risk Assessment of Veterinary Antibiotics and Heavy Metals in Typical Swine Wastewater Treatment Processes. East China Normal University, China (master thesis).
[4] Wan, L., 2016. Study on the New Technology of BCO + SBBR Aerobic Treatment for Scaled Pig Farms Wastewater (Biogas Slurry). Nanchang University (doctoral thesis).
[5] Zhang, Q.Q., Ying, G.G., Pan, C.G., Liu, Y.S., Zhao, J.L., 2015a. Comprehensive evaluation of antibiotics emission and fate in the river basins of China: source analysis, multimedia modeling, and linkage to bacterial resistance. Environ. Sci. Technol. 49, 6772 - 6782.
[6] Andrade T S, Henriques J F, Almeida A R, et al. Carbendazim exposure induces developmental, biochemical and behavioural disturbance in zebrafish embryos[J]. Aquatic Toxicology, 2016, 170: 390-399.
[7] Yu, Xiaoling, et al. "Tetracycline antibiotics as PI3K inhibitors in the Nrf2-mediated regulation of antioxidative stress in zebrafish larvae." Chemosphere 226 (2019): 696-703.
[8] Bownik, Adam, et al. "Procaine penicillin alters swimming behaviour and physiological parameters of Daphnia magna." Environmental Science and Pollution Research 26.18 (2019): 18662-18673.
Reviewer 3 Report
The paper is interesting but before publication on IJMS the Authors should also add a morphological approch (histological analysis or immunohistochemical reactions) in order to demonstrate the effects of erythromycin exposure on the muscle fibers of zebrafish and medaka .
Author Response
The paper is interesting but before publication on IJMS the Authors should also add a morphological approch (histological analysis or immunohistochemical reactions) in order to demonstrate the effects of erythromycin exposure on the muscle fibers of zebrafish and medaka.
Response 1: This paper focuses on the study of fish behavior changes under the influence of antibiotics. mRNA is expressed primarily to explain possible causes of behavioral changes, so we focused on genes that might influence behavior. Our analysis suggests that energy metabolism is one explanation. If muscle fibers are weakened, it may be a secondary cause of decreased swimming ability, but it's not our primary study. It is still valuable advice to understand more fully the changes in swimming ability.
Reviewer 4 Report
Li and Zhang examined acute erythromycin exposure to fish as a model of aquatic environmental impact for medicines released in waste water. This is an interesting set of experiments that examined swimming behavior and compared with muscle and brain gene expression. Overall, I have a very positive view of this submitted manuscript. There are some concerns about the presentation that I feel could be improved.
Major Points:
1. The levels of erythromycin were very high. All levels tested were higher than the levels known to have little or no effect. The levels were about 2,000 times that level, and this level was the main concentration that produced effects. This was the level used to do RNA-seq experiments. The levels used for these experiments should be better justified.
2. The description of Figure 2 was problematic. The reference to different parts of the figure were incorrect (the letters were wrong). The figure legend was incorrect, referring to letters a and b twice. It looks like the concentrations showing difference were incorrectly reported (2 and 20 ug/L rather than 200 ug/L and 2 mg/L). This must be remedied.
3. The lettering in Figures 4 and 6 are too small to be seen by readers. These figures should be improved.
4. The RNA-seq findings may point to changes in bioenergetics. It would be useful to stain mitochondria in the muscle and brain.
5. The brain RNA-seq shows phototransduction changes. Was the retina included with the brain sample, or how do you explain these findings? This should be clarified.
Author Response
- The levels of erythromycin were very high. All levels tested were higher than the levels known to have little or no effect. The levels were about 2,000 times that level, and this level was the main concentration that produced effects. This was the level used to do RNA-seq experiments. The levels used for these experiments should be better justified.
Response 1: Firstly, Antibiotic concentrations are relatively low in naturally healthy rivers. However, the antibiotic content is very high in the river reach where sewage is discharged from livestock farms or sewage treatment plants [1~3]. Secondly, The ecological environment of healthy rivers is stable and there are little problems that are harmful to aquatic life. This study focuses on the environmental hazards in polluted river sections. When studying the harm of antibiotics to aquatic organisms, high-concentration conditions were set to amplify the adverse effects to explore the mechanism, and are used in many experimental designs [4,5]. Finally, the detected concentrations range from ng L-1 to mg L-1 in the aquatic environment [6], so the experimental design concentration is in a reasonable range.
- The description of Figure 2 was problematic. The reference to different parts of the figure were incorrect (the letters were wrong). The figure legend was incorrect, referring to letters a and b twice. It looks like the concentrations showing difference were incorrectly reported (2 and 20 ug/L rather than 200 ug/L and 2 mg/L). This must be remedied.
Response 2: Legend errors have been corrected. The description of the 169line error has been modified.
- The lettering in Figures 4 and 6 are too small to be seen by readers. These figures should be improved.
Response 3: Figures 4 and 6 has been modified
- The RNA-seq findings may point to changes in bioenergetics. It would be useful to stain mitochondria in the muscle and brain.
Response 4: This experiment is a global preliminary exploration of the adverse effects of roxithromycin on fish, and we want to continue to perform specific analysis based on the results of this study. Not only the number of mitochondria in fish muscle, but also mitochondrial RNA expression under antibiotic stress
- The brain RNA-seq shows phototransduction changes. Was the retina included with the brain sample, or how do you explain these findings? This should be clarified.
Response 5: The description in the article is inaccurate, and the fish head was selected instead of the fish brain in this experiment. The reason is that in addition to the fish brain, many sensory organs in the fish head may be related to behavior. Modifications have been made in the modified version.
References:
[1] Ben, W., Qiang, Z., Adams, C., Zhang, H., Chen, L., 2008. Simultaneous determination of sulfonamides, tetracyclines and tiamulin in swine wastewater by solid-phase extraction and liquid chromatography-mass spectrometry. J. Chromatogr. A1202 (2), 173-180.
[2] Xu, Y.G., Yu, W.T., Ma, Q., Zhou, H., Jiang, C.M., 2015. The antibiotic in environment and its ecotoxicity: a review. Asian J. Ecotoxicol. 10 (3), 11-27.
[3] Lei, H., 2016. Occurrence, Fate and Environmental Risk Assessment of Veterinary Antibiotics and Heavy Metals in Typical Swine Wastewater Treatment Processes. East China Normal University, China (master thesis).
[4] Bownik, Adam, et al. "Procaine penicillin alters swimming behaviour and physiological parameters ofDaphnia magna." Environmental Science and Pollution Research 26.18 (2019): 18662-18673.
[5] Yu, Xiaoling, et al. "Tetracycline antibiotics as PI3K inhibitors in the Nrf2-mediated regulation of antioxidative stress in zebrafish larvae." Chemosphere 226 (2019): 696-703.
[6] Q. Zhang, A. Jia, Y. Wan, H. Liu, K. Wang, H. Peng, Z. Dong, J. Hu, Occurrences of three classes of antibiotics in a natural river basin: association with antibiotic-resistant Escherichia coli, Environ. Sci. Technol. 48(24) (2014) 14317.
Round 2
Reviewer 2 Report
Response 1 and 2 should be included in the introduction/discussion.
Response 5: since it is a very speculative and the authors stated "Because this study does not focus on lipid metabolism. Therefore, experiments were not designed from this perspective". Therefore more emphasis should be put on the fact that this is a pure author’s hypothesis
Author Response
Response 1 and 2 should be included in the introduction/discussion.
Thanks for the Suggestions of reviewers, we have added relevant contents to introduction (line 53 to 65).
Response 5: since it is a very speculative and the authors stated "Because this study does not focus on lipid metabolism. Therefore, experiments were not designed from this perspective". Therefore more emphasis should be put on the fact that this is a pure author’s hypothesis
Thanks for the Suggestions of reviewers, we have added relevant content to the discussion, which is detailed on line 342 to 344.
Reviewer 3 Report
The Authors should follow the suggestions. Therefore the results section should be improved.
Author Response
The Authors should follow the suggestions. Therefore the results section should be improved.
Thanks for your suggestion to improve our manuscript. Unfortunately, we are not allowed back to the laboratory because of the COVID-19. Thus, we are not able to work for the experiment currently. If the experiment was considered necessary, we are willing to supplement it after the lab is open. Thanks for your comments.
Reviewer 4 Report
In addition to the mitochondria numbers, the activity can be determine using specific dyes. However, I do not feel that this experiment is necessary for publication.Author Response
In addition to the mitochondria numbers, the activity can be determine using specific dyes. However, I do not feel that this experiment is necessary for publication.
Thank you very much for your Suggestions and tolerance. In the following research, we will add your valuable comments to the further experimental design. Thank you very much.